# Cancer cell sedimentation in 3D cultures reveals active migration regulated by self-generated gradients and adhesion sites

**Nikolaos M. Dimitriou**[1]*, **Salvador Flores-Torres**[1], **Maria Kyriakidou**[2], **Joseph Matthew Kinsella**[1], **Georgios D. Mitsis**[1]*

**1** Department of Bioengineering, McGill University, Montreal, QC, Canada, **2** Department of Human Genetics, McGill University, Montreal, QC, Canada

* nikolaos.dimitriou@mail.mcgill.ca (NMD); georgios.mitsis@mcgill.ca (GDM)

**Data Availability Statement:** All relevant data are within the manuscript and its Supporting information files. Links and accession numbers for the raw data and the code are provided in the

## Abstract

Cell sedimentation in 3D hydrogel cultures refers to the vertical migration of cells towards the bottom of the space. Understanding this poorly examined phenomenon may allow us to design better protocols to prevent it, as well as provide insights into the mechanobiology of cancer development. We conducted a multiscale experimental and mathematical examination of 3D cancer growth in triple negative breast cancer cells. Migration was examined in the presence and absence of Paclitaxel, in high and low adhesion environments and in the presence of fibroblasts. The observed behaviour was modeled by hypothesizing active migration due to self-generated chemotactic gradients. Our results did not reject this hypothesis, whereby migration was likely to be regulated by the MAPK and TGF-β pathways. The mathematical model enabled us to describe the experimental data in absence (normalized error<40%) and presence of Paclitaxel (normalized error<10%), suggesting inhibition of random motion and advection in the latter case. Inhibition of sedimentation in low adhesion and co-culture experiments further supported the conclusion that cells actively migrated downwards due to the presence of signals produced by cells already attached to the adhesive glass surface.

## Author summary

Cell sedimentation in 3D cultures can have both advantageous and unwanted effects. On the one side, the loss of the 3rd dimension is undesirable in a 3D culture. On the other side, it can reveal interesting migration phenomena given that passive mechanisms such as gravity are not potential contributors. In this work, we examined the phenomenon of sedimentation by combining 3D cultures and spatiotemporal mathematical modelling. We found that passive mechanisms such as gravity and matrix compression are not sufficient to bring the cells at the bottom. Our analysis showed that the cells actively migrate towards the bottom and in a collective manner, suggesting the presence of chemotactic gradients produced by cells already attached to the glass surface.

manuscript. Code repository: https://nmdimitriou.github.io/HyMetaGrowthXTreat/ Data repositories/ accession links: https://figshare.com/projects/3D-GROWTH-MDA-MB-231-SERIES-12/118989 https://www.ncbi.nlm.nih.gov/geo/query/acc.cgi?acc=GSE223350.

**Funding:** This study was supported by Stavros Niarchos Foundation Fellowship (F237055R00) (awarded to NMD) Werner Graupe (F202955R00) (awarded to NMD), McGill (90025) (awarded to both NMD, SFT), Compute Canada (RRG # 3975 awarded to both NMD, GDM), FRQNT (291010) (awarded to SFT), Cyprus Research and Innovation Foundation (Project: INTERNATIONAL/OTHER/0118/0018) (https://www.research.org.cy/en/) (awarded to GDM), and Natural Sciences and Engineering Research Council of Canada (Discovery grant 34362) (awarded to GDM). The funders had no role in the study design, data collection and analysis, or preparation of the manuscript.

**Competing interests:** The authors have declared that no competing interests exist.

# 1 Introduction

The use of 3D culture techniques in cancer research yields significant potential as these cultures have provided more realistic conditions of cancer growth in terms of morphology, gene expression, related biochemical processes, and altered drug toxicity compared to conventional 2D cultures. Current research on biomaterials [1, 2] on the surrounding environment of tumours, as well as bioprinting techniques [3] has enabled researchers to study various phenomena accompanying cancer growth, such as cellular aggregation, migration, and tissue expansion, along with the impact of the composition and geometry of the extracellular matrix (ECM) [4–7].

Furthermore, the *in silico* investigation of cancer growth using mathematical models allows for testing a wide array of scientific hypotheses at a considerably lower cost than biological experiments [8–10]. In this context, spatiotemporal models of cancer growth can be divided into three general categories; discrete (e.g., agent-based models), continuum (Partial Differential Equations, PDEs), and hybrid models [11, 12]. Discrete models can provide information on individual cell processes and tissue microarchitecture [13]. Continuum models have been widely used to describe the macroscopic aspects of tumour growth, albeit lacking experimental validation in many cases [14, 15]. More recently, they have been used to achieve a more detailed quantitative description of the macroscopic characteristics of spatiotemporal cancer growth and its response to therapy under both *in vitro* [16–19] and *in vivo* conditions [20–26]. Hybrid models attempt to provide a multiscale description of cancer growth by incorporating both continuous and discrete variables [27–29]. Tweedy et al. [30, 31] utilized experiments and hybrid discrete-continuum (HDC) models of chemotactic migration to investigate the role of self-generated chemotactic gradients in cancer migration.

Although there is growing literature on spatiotemporal models of cancer, their validation using experimental data is still relatively limited, yet important for quantitatively describing cancer-related mechanisms [32–40]. Model validation in cancer has previously been investigated in both forecasting [32, 41] and exploratory studies [42–48] of cancer growth using ODEs and PDEs. Recently, we also expanded validation to hybrid models using an integrated experimental and computational framework for the calibration and validation of hybrid models with 3D cell culture data using Bayesian inference and spatial statistical analysis techniques [49]. This framework enabled us to validate multiscale models using an efficient scale splitting technique that allows the preservation of data points for both calibration and validation.

In the present study, we focused on cell sedimentation in 3D cultures. We aimed to provide mechanistic explanations for its occurrence using an integrated multi-scale experimental and mathematical modelling approach. Sedimentation is defined as the phenomenon of cell migration towards the bottom of the culture space. It is generally undesirable because it alters the behaviour of the cells turning from 3D to 2D phenotypes [50], resulting in an early termination of the experiment or the need for alteration of hydrogel stiffness. The understanding of this phenomenon can potentially impact two major research fields. On the one hand, it can lead to better experimental design. In some cases, such as in spheroid formation assays, sedimentation can be an unwanted behavior [51], hence it is important to prevent it if possible. On the other hand, 3D migration assays can lead to important findings on the mechanobiology of cancer progression [6, 7, 16, 52, 53]. Another potential utility of sedimentation assays, is that they resemble a 3D analog of the 2D wound healing assays where the cells exhibit their migration potential by converging to the bottom instead of filling the gap in the 2D case. This phenomenon has been previously observed by [51] and interpreted as the result of cells moving towards the path of least resistance. In our recent study [49], we concluded that a self-generated chemotactic gradient mechanism of migration towards adhesion sites may also be

plausible. Here, we provide a comprehensive experimental and mathematical examination of this hypothesis in 3D cultures of triple-negative breast cancer (TNBC) cells. Specifically, we examined the role of active and passive migration mechanisms, their relation to the gene expression profiles of the cells using bulk RNA-seq data at various time-points and during paclitaxel treatment, the validity of the self-generated chemotactic gradient hypothesis, and the importance of adhesion sites.

## 2 Results

In this section, we first present the effect of Paclitaxel on the inhibition of migration as well as the transcriptomic profile of the cells. We then proceed to formulate the hypothesis of migration, and we examine its validity using Bayesian inference, spatial analysis, as well as data from further experiments including agarose coating of the glass bottom and the introduction of fibroblasts in the 3D culture.

### 2.1 Ruling out the possibility of gravity-induced migration

Cancer cell migration can be classified into two general categories; active and passive [54]. Passive migration is defined as the process in which cells migrate without the presence of active mechanisms, i.e. without mechanisms leading to the extension/contraction of the cell which eventually move the cell [54]. Initially, we hypothesized that gravity could be a major contributor to passive migration in our 3D experimental setup. To examine this hypothesis, we deactivated any potential active migration mechanisms by targeting the cell cytoskeleton with Paclitaxel. Paclitaxel has been considered as a migrastatic drug as it inhibits microtubule assembly, in turn, arresting cell division and migration [55]. The resulting cell distributions across different culture heights are shown in Fig 1b and A and B of S1 File. These show that

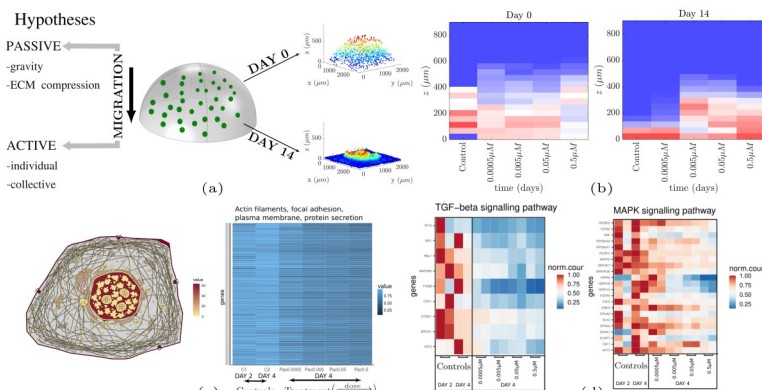

**Fig 1.** Investigation of migration mechanisms (a) Schematic representation of 3D cultures (center). The cells were uniformly distributed across the semi-ellipsoid on day 0 and actively migrated towards the glass bottom on day 14 (right panel). The examined hypotheses include passive and active migration mechanisms (left). (b) Cell distribution across different cell culture heights on days 0 and 14 in the presence (four different doses) and absence of Paclitaxel. Migration was inhibited for the three highest Paclitaxel doses, ruling out the possibility of passive migration. The glass bottom is located at z = 0μm. (c) (left) Cellular anatogram relating the gene expression activity to the corresponding cellular components. (right) Genes related to actin filaments, protein secretion, focal adhesion sites, and plasma membrane exhibited a monotonic decrease in their expression with respect to Paclitaxel dose. These cellular components are also involved in cell migration. (d) TGF-β (left) and MAPK (right) signalling pathways were downregulated as a function of the Paclitaxel dose. The two columns per condition belong to the two corresponding replicates. The inhibition of migration in combination with the downregulation of these pathways suggests that collective cell migration is more likely to occur than individual cell migration (i.e., epithelial-to-mesenchymal transition (EMT); Fig E of S1 File).

sedimentation was inhibited for the three highest administered doses compared to the controls. At the lowest dose, the cells continued to accumulate at the bottom of the space, likely to insufficient damage to the cytoskeleton. In conclusion, the cells of the treated samples maintained their initial position, suggesting that gravity was not the main reason for cell sedimentation.

## 2.2 ECM compression is not sufficient for cell sedimentation

Another possible reason that may lead to cell sedimentation is the collapse or gradual compression of the Matrigel ECM, which may occur over the course of the experiment. The ECM scaffolds in the treated and untreated samples had the same dimensions, and the initial number of cells was approximately the same. In presence of compression we would observe sedimentation in both treated and untreated samples. The observed cell height patterns in Fig 1b and A and B of S1 File suggest that ECM compression may have occurred at some extent; however, its magnitude was for sedimentation.

## 2.3 Transcriptomic analysis provides evidence of active migration

We investigated the presence of active migration by investigating the gene expression profiles of the cells in the control samples and samples treated with Paclitaxel. The bulk RNA was extracted at two time points (Days 2 and 4) for the untreated cells and at one time point (Day 4) for the treated cells. Analysis of the RNA-seq data (summarized in Fig C of S1 File, and Materials and methods Sections 5.4, 5.5, Section 1 of S1 File) suggests that the untreated samples exhibited upregulated expression levels of genes related to active migration, including actin filaments, focal adhesion sites, plasma membrane, and protein secretion. The cellular anatogram in Fig 1c (left) shows the expression of these genes at the corresponding sites of the cell. The heatmap of Fig 1c (right) shows that the expression levels of these genes remained highly similar between the untreated samples on days 2 and 4 (C1, C2 respectively) and tended to decrease as the Paclitaxel dose increased (from left to right).

## 2.4 Collective migration is a more plausible mechanism than individual migration

We next investigate the type of active migration occurring in untreated samples. Generally, active migration can be divided into two categories; individual and collective [56]. Individual cell migration is known to be regulated by epithelial-to-mesenchymal transition (EMT) [56], whereas collective migration is a more complex process that involves molecular mechanisms related to adherens junctions (AJ), MAP kinase (MAPK), and TGF-β signalling pathways [57]. To elucidate the migration type, we isolated the expression levels of genes related to individual (EMT) and collective (AJ, MAPK, and TGF-β) migration. The results presented in Fig E of S1 File show no distinct pattern for EMT between treated and untreated samples, suggesting that the EMT pathway was not affected by Paclitaxel. However, we observed the upregulation of protein-coding genes involved in MAPK and TGF-β (Fig 1d) and over-representation of the AJ (Fig D of S1 File) in the untreated samples, suggesting the presence of collective migration.

## 2.5 The role of the glass surface and a mechanistic interpretation of cell migration

Based on results described above, cell sedimentation is more likely to be attributed to active and collective migration. Recently, Friedl et al. [57] reported that cells migrating collectively can be separated into leaders and followers. Leader cells orient and move based on stimuli

such as soluble factors. An example of soluble factor diffusing in the extracellular space can be TGF-β although it was not examined in this study. These stimuli then induce MAPK signaling and downstream Rac1 for protrusion formation and direction sensing. Leader cell polarity is further supported by AJ signaling, which controls leader cell polarization and anterior protrusion.

Acknowledging that leader cells move based on signals, it is worth questioning the source of these signals. It should be noted that MDA-MB-231 cells are naturally adherent [58], and they were initially uniformly distributed in the 3D space, i.e., some of them are close to the glass bottom space. Based on these observations, we hypothesized that, first, the cells closer to the bottom attach to the glass, which is a favorable space for colony formation owing to its adhesiveness. Second, the cells attached to the glass secrete signals to stimulate cell migration and aggregation due to their sparse distribution in the space. Third, these signals diffuse in 3D space, forming a gradient from the bottom to the top of the space. Fourth, the cells floating in the 3D ECM may have already formed clusters, due to, for example, cell division or approaching their closest neighbours. These clusters sense the gradient and organize themselves into leaders and followers. Finally, the clusters move towards the direction of the increase in the signal density. Thus, we attributed this sedimentation phenomenon to a self-generated chemoattractant gradient produced by the cells to indicate a favorable site of colonization, as summarized in Fig 2.

## 2.6 Mathematical modeling of cellular dynamics

We formulated the active migration hypothesis described above into a hybrid (discrete-continuum) 2-level spatiotemporal model. The model quantifies the spatial cancer cell density

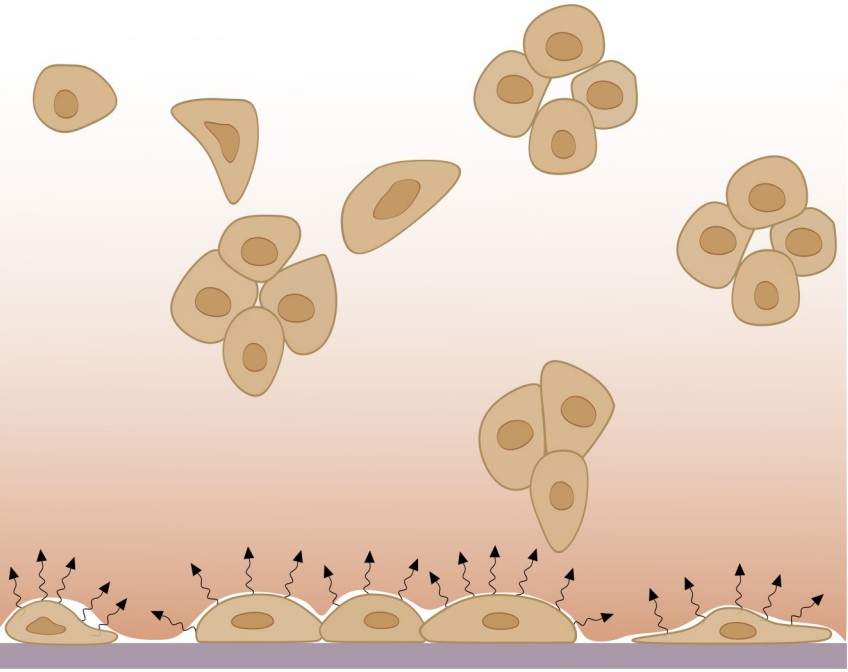

**Fig 2. Schematic representation of the hypothesis of signal induced migration towards the bottom of the space.** Cells from the initial stages of the experiment that are close to the bottom attach to it and secrete signals to stimulate aggregation. Signal diffusion in the 3D space forms a gradient that decreases from the bottom to the top. Floating cells orient and migrate towards the direction of increased signal concentration.

profiles, $u$, and chemoattractant factors, $f$, using a system of PDEs and a cellular automaton (Material and methods Section 5.8, Section 2 of S1 File). It takes into account the random motion of cancer cells and chemotactic signals, growth of cancer cell density, and migration of cells towards the direction in which the signals increase. For simplicity, we assumed that the gradient was formed during the initial stages of development; hence, we implemented a gradient, decreasing from bottom to top, as the initial signal concentration profile.

To examine the validity of the proposed model, we initially estimated the parameters of the continuum model with respect to experimental data using Bayesian inference (Materials and methods Section 5.9, Section 3 of S1 File). For a given parameter set, the resulting cell density profiles were compared with the *in vitro* estimated cell density profiles of a dataset. This process was applied to each of the 32 datasets separately (12 untreated and 5 datasets for each administered dose of Paclitaxel). Each dataset consisted of seven samples on days 0, 2, 5, 7, 9, 12, and 14. Approximately 20000 different sets of model parameters were assessed using the Transitional Markov Chain Monte Carlo (TMCMC) method for each of the 32 datasets (Section 3 of S1 File, Tables A-E and Fig I of S1 File). The parameters affected by Paclitaxel were mostly $k$, $D_u$, $\chi$. The first parameter is related to cell death, and the rest are related to the kinetics of the cell, specifically random motion and biased movement. We assumed that the parameters $s$, $D_f$ which represent the growth rate of the cells and the diffusivity of the chemotactic signals, respectively, were not affected by the treatment. The former, $s$, was offset by the cell death rate, $k$, and the latter because of the assumption that pre-existing signals were not affected by the treatment. The affected parameters were calibrated within broad boundaries, whereas the unaffected parameters were calibrated within the average ± standard deviation obtained from controls. The resulting marginal posterior distributions of the model parameters are presented in Fig 3a. The affected parameters for cell movement ($D_u$, $\chi$) tended to decrease as the dose increased, while the cell death rate ($k$) increased with respect to the dose.

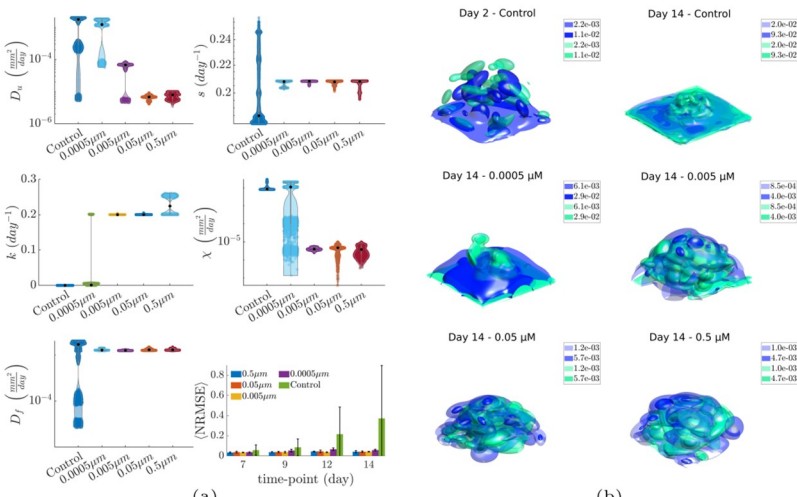

**Fig 3. Continuum model calibration.** (a) Posterior marginal distributions of the estimated model parameters for the examined experimental conditions. The NRMSE values for these conditions (bottom right) suggest overall low errors for all conditions, with the model yielding overall higher errors for the non-treatment condition during cell aggregation. (b) Surface plots of the *in vitro* and *in silico* density profiles for representative datasets on days 2 and 14. Paclitaxel was administered on day 5; hence, day 2 was not affected by treatment, while on day 14, we observed a difference in the spatial cell density distributions across the non-treatment and treatment conditions. The distributions remained relatively constant for the three highest doses because cell migration was inhibited, while for the lowest dose and control conditions. the distribution changed due to cell sedimentation. The green coloured profiles correspond to experiments and the blue to simulations.

The density plots of Fig 3b and J of S1 File) show an overall agreement between experimental data and simulations obtained from the calibrated model. The calculated Normalized Root Mean Squared Error (NRMSE) (Section 4 of S1 File) for the most probable values of the parameters (Fig 3, bottom right) suggested an NRMSE value around 5% for each time point for the treatment datasets. The NRMSE values were higher for the later stages of the control datasets; however, they remained within reasonable levels.

The estimated model parameters were subsequently used in the hybrid model (Fig 4) for each dataset. The resulting *in silico* cellular coordinates were analysed and compared with the corresponding *in vitro* coordinates of the centroids from the segmented fluorescent nuclei of the cells. The *in silico* cells reproduced the overall behaviour of the *in vitro* cells for each of the examined conditions in terms of both spatial profiles (Fig 4a and 4c), and longitudinal

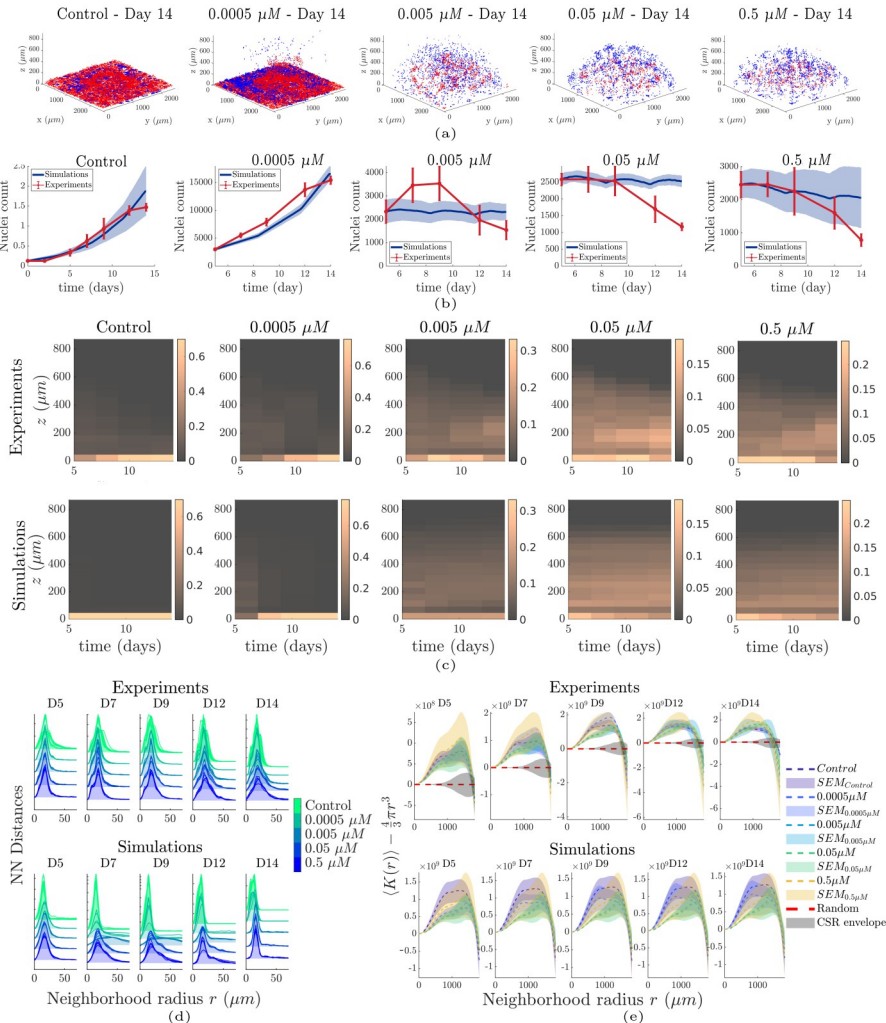

**Fig 4. Hybrid modelling results and spatial analysis.** (a) 3D plots of cell locations on day 14 for the control and treatment conditions. Red: experiments/Blue: simulations. (b) Predicted and experimentally observed cell numbers. The total cell population tended to decrease with increasing doses. (c) Histograms of the proportion of cells as a function of position in the culture across time for the experimental (top row) and simulated (bottom row) data, respectively. (d) Nearest Neighbour distances across the examined conditions with respect to time, for the experimental and simulated data. (e) Complete Spatial Randomness test; average values of the $K$-function across all samples and the corresponding standard error of mean (SEM).

population size (Fig 4b). The quantitative characterization of their spatial distributions was performed using their inter-nucleic (IN) and nearest neighbor (NN) Euclidean distances, as well as the Complete Spatial Randomness (CSR) test using Ripley's *K*-function. The IN distances were stable across samples and time, with a characteristic peak at $\sim 1mm$ (Fig L of S1 File). The NN distances were initially widely distributed in the control samples (Fig M of S1 File), and tended to become narrower at later stages with a characteristic peak found at $\sim 15\mu m$, which is approximately equal to the cell size. As the treatment dose increased, the NN distribution became wider (Fig 4d), indicating that the spatial distributions became sparser. Similar results were obtained from the simulated data. However, the width of the distributions produced by the simulated data remained similar across the different doses and the characteristic peak was shifted towards larger values. This also indicates an increase in the sparsity of the spatial distributions, but in a more uniform manner compared to the experimental observations. The CSR test shows an upward separation from the theoretical uniform random distribution, indicating the presence of clustered patterns (Fig 4e). The control samples yielded clustered patterns that became more pronounced with respect to time (Section 5, Fig K a of S1 File), possibly due to cell aggregation at the bottom. The clustered patterns in the treatment datasets became more pronounced with increasing dose and time. Additionally, a more pronounced dispersion for larger neighbourhood radii was observed in these data with respect to both dose and time. Combining these two observations, we conclude that these patterns were possibly due to the radial shrinkage of the spatial cell distributions towards the centroid of the scaffold.

## 2.7 The effect of the glass on cell migration

To further examine the validity of our cell migration hypothesis, we performed an additional 3D cell culture experiment in which we coated the glass bottom with a non-adhesive material. Specifically, we applied agarose coating on the surface of the glass and deposited the cell/Matrigel mixture in pockets of air on the surface of the agarose (Fig 5a). In total, ten samples were created and tracked on days 0 and 14. The results shown in Fig 5b show that the cells overall maintained their position as compared to the results obtained from the control experiment shown in Fig 5c. Additionally, we observed reduced cell viability compared to the control experiment (Fig G of S1 File), which could be attributed to the absence of adhesiveness in the surrounding environment.

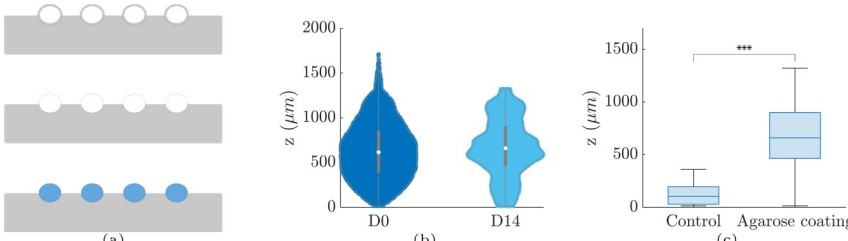

**Fig 5. Coating of the glass bottom with agarose.** (a) Schematic representation of the experimental procedure. The gray area represents the agarose deposited on the glass surface. The white disks denote the air bubbles injected to create pockets, the second row represents the removal of the inflated agarose surface, and the blue disks represent the cell/Matrigel mixture deposited on the pockets of the polymerized agarose. (b) Violin plots of the distribution of cells across the z-dimension of the cell culture for days 0 and 14 of the agarose coating experiment. Cell positions were overall maintained throughout the course of the experiment. (c) Boxplots of the cell distributions across the z-dimension of the control and the agarose coating experiments for day 14. The three asterisks denote p-value $< 10^{-3}$ that was calculated using the Kruskal-Wallis test between the two distributions.

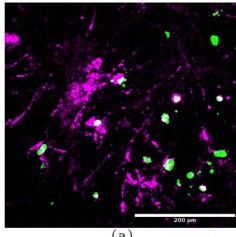
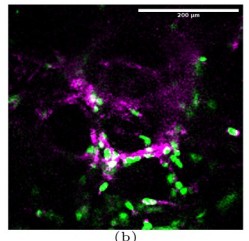
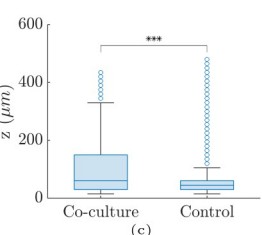

**Fig 6. TNBC cells and fibroblasts co-culture.** Fibroblasts were suspended in the area surrounding the Matrigel. Cancer cells (green) and fibroblasts (magenta) were mixed at (a) the bottom of the space and (b) at 195 μm height. (c) Boxplots of cancer cell distribution across different culture heights for the co-culture and cancer cell monoculture for day 14. The three asterisks denote p-value $< 10^{-3}$ that was calculated using the Kruskal-Wallis test between the two distributions.

## 2.8 Fibroblasts surrounding the Matrigel scaffold antagonize biased migration signalling

To further examine the hypothesis of signal-induced migration, we performed an additional experiment that included fibroblasts. Specifically, we suspended fibroblasts in the area surrounding the Matrigel scaffolds. Over time, the fibroblasts attached to the periphery of the Matrigel scaffolds and exhibited some level of invasion in the Matrigel (Fig 6a and 6b). The hypothesis behind this experiment is that fibroblasts would induce signals that would bias the migration of cancer cells towards them; in turn, some of the floating cells would move and eventually mix with the fibroblasts that surrounded the scaffolds, while others would move towards the bottom. The results shown in Fig 6c confirm this hypothesis and showed that cancer cells maintained elevated positions in the presence of fibroblasts, as compared to the controls. Thus, the adhesion of the fibroblasts to the cancer cells is likely the reason that prevented downward migration, which was preceded by the movement of the fibroblasts toward the cancer cells, likely due to signaling.

## 3 Discussion

We used a combination of 3D cultures and mathematical modeling to characterize cell sedimentation. Our results suggest that this somewhat unexpected behavior is due to active migration regulated by signalling gradients created by the cells that were already attached to the bottom.

### 3.1 3D cell culture system, migration inhibition, and molecular mechanisms

Somewhat surprisingly, there is limited literature discussing the phenomenon of sedimentation. In a recent study, Liu et al. [51] observed the same phenomenon, interpreting it as the result of cells moving towards the path of least resistance. In ECM, the path of least resistance can be interpreted as a path with pores large enough to allow a cell to move without the need for compression [59]. Here, we should note that we do not expect major differences in the pore size across the 3D scaffold that would bias cell migration in a specific direction. Additionally, the assumption made by Liu et al. [51] is in disagreement with other observations showing that cells tend to move towards the direction of greater matrix stiffness, termed durotactic movement [60], which may also be combined with chemotaxis in the examined setup. Nevertheless, we expected a local change in ECM stiffness induced by cell contractility [7]; however,

this change is expected to be restored at approximately 20 μm distance from an MDA-MB-231 cell in a Matrigel ECM based on [7]. Additional experiments have shown that cells can polarize and migrate based on the alignment of ECM fibers [5], a movement referred to as contact guidance. However, we did not expect any strong fiber alignment that can influence cell migration, as Matrigel fibers tend to polymerize in random directions [61].

To determine the presence of active or passive migration, we repeated the same experimental procedure and introduced Paclitaxel on day 5, when accumulation started to become apparent. Studies have classified Paclitaxel as a migration inhibitor [55] as it inhibits microtubule assembly, subsequently preventing cell division and migration. The administered doses did not eliminate the cells over the experiment time course, while three out of four doses inhibited cell migration and cell accumulation at the bottom. Therefore, we ruled out the possibility of gravity-induced migration or accumulation due to ECM compression. Additionally, to further test the presence of active migration, we performed a bulk RNA-seq experiment using the same experimental setup and compared the expression profiles between non-treated and Paclitaxel-treated samples. The results revealed an increased expression of genes related to the mechanisms of active migration in the non-treated samples. Specifically, we found that cells were more likely to migrate collectively than individually.

## 3.2 Modelling cancer growth and aggregation at the bottom of the space

The evidence on collective migration led us to hypothesize that cells move based on cues that specifically originate from the bottom of the space. Since the glass surface cannot secrete signals, we hypothesized that the source of these signals was the cells attached to the bottom, and the result of this secretion was to stimulate aggregation. To examine the biophysical mechanisms of spatiotemporal cancer development and cell sedimentation, we formulated this hypothesis using a hybrid discrete-continuum mathematical model. The calibration process was performed on the continuum part of the model using Bayesian inference and the TMCMC algorithm. The results showed an overall good agreement between the experiments and simulations (Fig 3), with the simulation results reproducing cell aggregation at the bottom under non-treatment conditions, cell death, and migration inhibition in the presence of treatment. Nevertheless, the increased range of the posterior distributions suggested identifiability issues, possibly owing to misaligned 3D culture samples across different time-points, small geometric differences across scaffolds, as well as limited number of time-points. The examination of different models could allow us to find a model with better balance between complexity and fitness, however model selection was not performed due to limited computational power. The calibrated parameters were then transferred to the full HDC model, which provided details regarding the location of the cells. The results of the HDC model were validated using spatial point-pattern analysis techniques, which revealed information on the spatial organization of cells in both experiments and simulations.

## 3.3 Other possible mechanisms of migration

The differential gene expression analysis provided evidence of collective migration that are likely to performed using the leader-follower mechanism. Nevertheless, it cannot provide sufficient evidence of its presence as there is a lack of dense time-lapse microscopy data that focus on the required proteins. Thus, we examined the presence of other types of migration. Some candidates are sprouting and branching [62], multicellular streaming [62], contact inhibition of locomotion [63, 64], and supracellular contraction [65] (Fig F of S1 File). Among these mechanisms, we found that contact inhibition of locomotion can contribute to cell migration

as Rhoa and Rac1, which are responsible for cell polarity and participate in this type of migration [63, 64], exhibited high levels of expression in the control samples.

## 3.4 Spatial patterns and mechanisms

Spatial point-pattern analysis revealed clustered patterns across the examined conditions that became more pronounced with respect to time (Fig 4e). These patterns, even though they appeared similar, were the result of different mechanisms. First, we should note that these clustered patterns appeared because of the initial culture geometry. For the control datasets, more pronounced clustering across time was observed mainly because of the biased migration that resulted in cell aggregation at the bottom. The mechanism underlying this process is the advection term. As shown in Fig N of S1 File, we demonstrated that a set of parameters corresponding to high diffusion and low advection produces less pronounced clustered patterns compared to a set corresponding to low diffusion and high advection. However, the pronounced clustering observed in the treatment data was not due to migration, but due to the shrinkage of the spatial distributions. Specifically, for the experiments, we found that the clustered patterns became more pronounced with respect to both time and dosage. This result contradicts the simulation results, which showed the opposite. This led us to conclude that pronounced clustering may have emerged from radial cell death in the spatial distributions, possibly due to diffusion of the drug in the Matrigel that may have produced concentration gradients. In contrast, the model assumed a uniform concentration of the drug across the space; hence, there was no cell-killing bias with respect to the location of the cells in the space. Additional simulations using a radius-dependent drug concentration confirmed that clustering increased in the presence of a drug concentration gradient (Section 5, Fig K of S1 File).

## 3.5 The importance of attachment sites in cell growth and migration

The examined hypothesis states that cells closer to the glass bottom attach to it and secrete signals enabling floating cells to migrate. To further examine this hypothesis, we performed additional experiments in which we covered the glass bottom with a non-adhesive material (agarose). Agarose is a biocompatible biopolymer with a poor ability to promote cell attachment owing to its low adhesiveness [66]. The results shown in Fig 5 showed that the cells tended to maintain their initial positions, confirming the hypothesis that the adhesive properties of the glass bottom affected cell migration. Additionally, we observed reduced cell viability in the coated samples, which can be attributed to the low degree of attachment between the cells and the surrounding material. Other studies have also shown that monolayers cultured in agarose-treated plates exhibit reduced proliferation or viability compared to other materials such as Matrigel [67, 68]. This in turn implies that cell clustering is important to maintain cell viability and that cell-surface attachment is also a major contributor in cell viability.

In contrast with the previous experiment, the presence of fibroblasts likely induced migration signals and increased adhesion leading to cell-fibroblast mixing across the outer surface of the Matrigel scaffolds. In turn, the cancer cells exhibited less pronounced migration towards the bottom and maintained elevated positions in the 3D space (Fig 6). At this point we should not exclude the possibility that fibroblasts physically blocked cell migration, however their aggregation with the cancer cells can be due to signalling. Other studies have also shown that the presence of fibroblasts in 3D cancer cell cultures results in more compact spheroids as well as altered intercellular signalling and gene expression, leading to altered cell proliferation and migration [50, 69–71].

### 3.6 Semi-ellipsoids as a 3D migration assay

Over the years, wound healing assays have become the standard protocol for examining *in vitro* cell kinetics under certain conditions. Wound healing assays typically include monolayers of adherent cells in which a cell-free gap is created, and the gap is closed by the adjacent cells. The current experimental setup presents many similarities to the wound healing assay, except that cells are initially cultured in 3D and migrate towards the bottom. In addition, in the present study, we showed that under certain conditions, for example, the presence of a *migrastatic* drug or non-adhesive surface the cells stopped migrating. This would also be the case for conventional wound-healing assays. However, our 3D experimental setup presents some advantages compared with the wound healing assay. First, the introduction of the 3rd dimension may present significant alterations in the behaviour of the cells, such as cytotoxicity, growth, and related biochemical processes, while the histological and molecular features of *in vitro* 3D spheroids exhibit more similarities with xenografts than conventional 2D monolayers [50, 72–75]. Second, the initial uniform and sparse distribution of the cells in 3D space allows not only the observation of sedimentation, but also additional phenomena that occur during the culture, such as cell division and cluster formation. In addition to these advantages, the setup is easy to construct, the experimental process is relatively short, and allows for a large number of replicates—in the present study we generated 5 and 12 replicates per time-point, for each experiment. More importantly, quantification of cell behaviour can be achieved with the use of mathematical models and spatial analysis techniques, as done herein. In conclusion, the presented methodology presents many advances compared to standard monolayer protocols, and can be expanded to the study of many different biological mechanisms.

## 4 Conclusion

We examined cell sedimentation in 3D cancer cell cultures. The introduction of the *migrastatic* drug Paclitaxel inhibited this sedimentation, suggesting that active migration mechanisms are involved in sedimentation. Furthermore, RNA-seq analysis showed that cancer cells were more likely to migrate towards the bottom in a collective manner than individually. Based on this evidence, we formulated a hypothesis stating that cells floating in 3D migrate due to signals produced by cells already located at the bottom of the space. Mathematical modelling of this hypothesis validated the experimental observations and provided insights into the mechanisms and spatial organization of the cells. To further validate this hypothesis of aggregation due to the presence of adhesive areas, we coated the glass bottom with agarose to prevent cell adhesion, resulting in the absence of sedimentation. Additionally, to further validate the hypothesis of migration due to signaling, we introduced fibroblasts to the surrounding area of the Matrigel scaffold, which resulted in both sedimentation and maintenance of clusters in 3D space. Overall, the examined 3D culture setting provided important insights into the behaviour of cancer cells, and revealed that TNBC cells tend to move towards areas of increased adhesion. The proposed mathematical modelling approach enabled us to characterize the mechanisms and spatial organization of cell growth. Future extensions of the present work may include the expansion of this framework to other types of cells, e.g. cancer cells co-cultured with immune cells, and the examination of different drug/dose schemes for the optimization of drug efficacy in terms of both cell growth and migration, as well as the effect of drug resistance and its relationship with cell migration.

## 5 Materials & methods

To examine the phenomenon of sedimentation in 3D cultures, we performed the following experimental and computational modelling steps. First, we performed 3D Matrigel cultures of TNBC cells and we collected spatial data using confocal microscopy across different time-

points. To examine the effect of passive migration we performed 3D matrigel cell cultures treated with Paclitaxel, which affects the microtubules of the cells, hence inhibiting any possible active migration. Upon rejection of passive migration mechanisms, we performed RNA-seq experiment and differential gene expression analysis to examine the presence of mechanisms responsible of migration. These observations led to the development of a hypothesis of cell sedimentation involving self-generated chemotactic gradient which was examined using a hybrid discrete-continuum mathematical model. The model was calibrated and validated with the experimental data. Finally, to further validate the examined hypothesis we performed two additional experiments that involved cancer cell/fibroblast co-cultures and 3D cultures in agarose coated bottom. These experiment examined the signalling hypothesis and the contribution of sites of high adhesion to the migration. The methods are described in the following paragraphs.

## 5.1 Cell preparation

Triple-negative breast cancer (TNBC) cells from the MDA-MB-231 cell line (ATCC) with nuclear GFP (histone transfection) were thawed and cultured at 5% CO2 and 37˚C in DMEM (Gibco) at pH 7.2 supplemented with 10% fetal bovine serum (Wisent Bioproducts), 100 U/mL penicillin, 100 μg/mL streptomycin, and 0.25 μg/mL amphotericin B (Sigma) in T-75 flasks (Corning). The cells were passaged before reaching 85% confluence. Three passages were performed before the 3D cultures; cells were rinsed twice with DPBS and trypsin-EDTA (0.25%-1X, Gibco) was used to harvest them. The MDA-MB-231 cell line was validated with short tandem repeat (STR) analysis [2].

## 5.2 3D cell cultures

A cell-Matrigel (Corning) suspension was created using 2 mL of Matrigel (4˚C) and $5 \times 10^4$ MDA-MB-231/GFP cells. Droplets of a 5 μL cell-Matrigel mixture were manually deposited onto a high-performance #1.5 glass-bottom 6-well plate (0.170±0.005 mm) (Fisher Scientific). The datasets were separated into non-treatment and treatment groups. The datasets representing treatment conditions were administered with Paclitaxel on the 5th day of the experiment. The administered doses were 0.5 μM, 0.05 μM, 0.005 μM, and 0.0005 μM. In total, 32 datasets were generated, including 12 biological control replicates and 5 biological replicates for each dose. Each of these samples were monitored using confocal microscopy on days 0, 2, 5, 7, 9, 12, and 14. To assess the reproducibility of our results, we performed the control experiments in two independent sessions, and in both sessions sedimentation occurred. The data are publicly available in the corresponding FigShare repository.

## 5.3 Imaging and data preparation

Data acquisition was performed every 2–3 days for a total of 15 days using a confocal microscope (Nikon A1R HD25), coupled with a cell culture chamber. The dimensions of the 3D cultures were approximately $2.5 \times 2.5 \times 0.9$ mm$^3$. Cell localization was made possible by the presence of GFP fluorophore in the cell nuclei. Fluorescent nuclei were segmented using the image processing and segmentation pipeline described in [76]. The pipeline was implemented in MATLAB [77] and imageJ [78] and is publicly available in the GitHub repository of the present study. The segmented nuclei were then mapped to a 3D Cartesian space by detecting their centroid locations using a 26-connected neighbourhood tracing algorithm implemented in MATLAB [77]. The final step was the calculation of the spatial density profiles of the cells represented by their centroids using Kernel Density estimation via the diffusion method [79]. Density calculations were performed using a grid of size $167 \times 167 \times 61$, such that each cell

occupied approximately one grid point. The density matrices were linearly interpolated to match the spatial grid size of the simulations ($480 \times 480 \times 176$).

## 5.4 RNA-seq experiment

We repeated the same experimental procedure described in Materials and Methods sections A, B, increasing the number of cells and the Matrigel volume to achieve the threshold of 500k cells required for the sequencer. The following protocol was implemented for each cell/Matrigel sample. Fresh cell culture media were warmed at 37˚C and supplemented with collagenase and dispase 1X (Sigma Aldrich, Cat. # 11097113001). Micropipettes were used to break down the cell-laden Matrigel constructs using suction force, and these were incubated for 30 min in a cell culture incubator (95% relative humidity, 37˚C, and 5% $CO_2$). The cells were centrifuged at 300 g for 5 min and the supernatant was discarded. The cells were resuspended in TRIzol (Fisher Scientific) with a 3:1 ratio of sample volume. The samples were frozen at -80˚C. The RNeasy Plus Universal Mini Kit (QIAGEN) was used to extract total RNA from the QIA cube Connect, according to the manufacturer's instructions. Library preparation construction and sequencing were carried out at Génome Québec using the Illumina NovaSeq 6000 platform, and paired-end reads (PE100) were produced. The RNA-seq experiment consisted of 24 samples in total. For the non-treatment conditions, the samples were extracted from days 2 and 4 with 2 biological replicates for each time-point. For each dose, the samples were isolated in day 4 with 2 biological replicates for each dose (0.5 μM, 0.05 μM, 0.005 μM, and 0.0005 μM), respectively. The data are publicly available in Gene Expression Omnibus (GEO) at GSE223350.

## 5.5 Transcriptomic analysis

The pipeline for the transcriptomic analysis is presented in Fig C of S1 File. Adapter sequences and low quality bases were removed using Trimmomatic (v.0.39) [80]. Reads were scanned and truncated when the average quality of the three-nucleotide sliding window fell below a threshold of 3. Short reads were discarded after trimming (<36 bp). Quality control metrics were obtained using FASTQC (v.0.11.9) [81] and SAMtools (v.1.10) [82]. The reads were aligned to the reference genome hg19 (GRCh37) [83] using STAR (v.2.7.5) [84], and only uniquely aligned reads were retained. Gene expression levels were calculated by quantifying uniquely mapped reads mapped to exonic regions (n = 63,568 genes) using Rsubread (v.2.4.2) [85]. Both normalization and differential expression analyses were performed using DESeq2 (v.1.30.1) [86]. The results obtained from the differential gene expression analysis across treatment and non-treatment conditions were filtered using the independent hypothesis weighting method (v.1.18.0) [87]. Gene Ontology over-representation tests and gene set enrichment analysis were performed using ClusterProfiler (v.3.18.1) [88]. Finally, gene expression localization was performed using OncoEnrichR (v.1.4.0) [89] and the ComPPI database (v.2.1.1) [90]. The cellular anatogram in Fig 1c, was generated using gganatogram (v.1.1.1) [91]. The code for the transcriptomic analysis can be found in the corresponding GitHub repository.

## 5.6 Agarose coating and 3D cell culture experiment

As described in [49], agarose 4% (w/v) was heated to boiling point, and liquid agarose (2.5mL) was used to cover the bottom of a high-performance #1.5 6-well plate (Fisher Scientific). This process was repeated for all five wells. While the agarose was still in the liquid phase, we created 5μL of air bubbles on the surface of the agarose using micropipettes. Upon polymerization of agarose, we discarded the outer layer of the bubble, creating a pocket. Finally, each pocket was filled with 5μL of the cell/Matrigel mixture described in Materials and Methods Section

5.2. Data were acquired on days 0, 2, 7 and 14 using a confocal microscope (Nikon A1RHD25).

## 5.7 TNBC/cancer-associated fibroblast co-cultures

TNBC cells of the MDA-MB-231/GFP cell line were suspended in Matrigel, according to the protocol described in Materials and Methods sections A and B. Cancer-associated fibroblasts of the cell line IMR-90 (ATCC) labelled with mCherry were introduced to the adjacent space of the cell culture space and settled on the surface surrounding the Matrigel scaffold. Throughout the duration of the experiment (14 days), the fibroblasts invaded the Matrigel scaffold and mixed with cancer cells. Data were acquired on days 0 and 14 using a confocal microscope (Nikon A1RHD25).

## 5.8 Mathematical model

Similar to our previous study on the validation of models using 3D cell culture data [49], we used a system of two Keller-Segel (KS) type equations for cancer cell density and chemotactic agent density. The spatiotemporal evolution of the cancer cell, $u$, and chemotactic agent, $f$, densities were obtained by the following PDEs:

$$\frac{\partial u}{\partial t} = D_u \nabla^2 u + su(1-u) - ku - \chi \nabla \cdot [u(1-u)\nabla f], \text{ in } \Omega \tag{1}$$

$$\frac{\partial f}{\partial t} = D_f \nabla^2 f, \text{ in } \Omega \tag{2}$$

$$\nabla u \cdot \vec{n} = \nabla f \cdot \vec{n} = 0, \text{ in } \partial\Omega \tag{3}$$

$$u(x, y, z, t = 0) = \text{observed} \tag{4}$$

$$f(x, y, z, t = 0) = e^{\left(-\frac{z}{0.26 \text{ (mm)}}\right)} \text{I}, \text{ I} = \begin{cases} 1, & \text{if } u > 0 \\ 0, & \text{if } u = 0 \end{cases} \tag{5}$$

where $D_u$, $D_f$ are the diffusion constants, $s$ is the growth constant of the cell density, $k$ is the cell death rate in the presence of treatment, and $\chi$ is the advection constant of the cells. The right-hand side consists of the diffusion terms $D_u \nabla^2 u$, $D_f \nabla^2 f$, which represent the random motion of the cancer cells and signals, respectively; the growth term $su(1-u)$, which increases the density of the tumour in a logistic manner; $ku$ is the cell death term in the presence of treatment; and the nonlinear advection term $-\chi \nabla \cdot [u(1-u)\nabla f]$ which represents the biased movement of the cells towards the direction where the gradient of the chemotactic signal density increases. According to the hypothesis of cell aggregation at the bottom (Results E), cells at the bottom secrete signals to stimulate cell migration and aggregation, and these signals diffuse in the 3D space, forming a gradient from the bottom to the top. Since cell attachment at the bottom was observed during the initial stages of the experiment, we assumed that signal secretion and diffusion occurred at the beginning of the experiment. For simplicity, we incorporated a signal gradient that decreased exponentially from bottom to top in the Initial Conditions (IC), as shown in Eq (5). In our previous analysis [49], we found that the signal production term in Eq (2) was insensitive to the produced output; hence, we excluded it from the current model.

We hybridized the KS model using the technique presented in [92, 93]. Specifically, we discretized Eq (1) using the forward time central differences scheme (FTCS) and the

approximations found in [94]:

$$
\begin{aligned}
u_{i,j,k}^{n+1} &= u_{i,j,k}^{n}P_0 + u_{i+1,j,k}^{n}P_1 + u_{i-1,j,k}^{n}P_2 + u_{i,j+1,k}^{n}P_3 \\
&\quad + u_{i,j-1,k}^{n}P_4 + u_{i,j,k+1}^{n}P_5 + u_{i,j,k-1}^{n}P_6
\end{aligned}
\tag{6}
$$

where the grouped terms $P_i$, $i = 0, \ldots, 6$ denote the probabilities of the cells of remaining stationary ($P_0$) or moving back ($P_1$), front ($P_2$), left ($P_3$), right ($P_4$), down ($P_5$), up ($P_6$), defined as

$$
\begin{aligned}
P_0 &= 1 - \frac{6D_u dt}{dx^2} \\[6pt]
P_{1,2} &= \frac{D_u dt}{dx^2} \mp \frac{\chi dt}{4dx^2}(f_{i+1,j,k} - f_{i-1,j,k}) \\[6pt]
P_{3,4} &= \frac{D_u dt}{dx^2} \mp \frac{\chi dt}{4dx^2}(f_{i,j+1,k} - f_{i,j-1,k}) \\[6pt]
P_{5,6} &= \frac{D_u dt}{dx^2} \mp \frac{\chi dt}{4dx^2}(f_{i,j,k+1} - f_{i,j,k-1})
\end{aligned}
\tag{7}
$$

Since the cells were approximately 15 μm in size and the spatial grid points were 5.2 μm apart, we assumed that each cell occupied three grid points in each direction. To account for this, we modified Eqs (6) and (7) by changing the indices that point in a direction to two grid points instead of one, that is, i±3 instead of i±1. The movement probabilities were then passed to a cellular automaton that updated the position and state of each cell.

The cellular automaton (CA) is shown in Fig H a of S1 File. The CA takes into account three cellular states; alive, quiescent, and dead. At every time step, it checks whether a cell can undergo spontaneous death, based on the probabilities shown in Fig H b of S1 File, and updates the age of the alive cells. The probability of spontaneous death increased after Day 10. This hypothesis is based on increased cell crowding, which results in a potential shortage of nutrients or accumulation of metabolic waste products. The CA checks whether any cell has reached the proliferation age determined based on the estimated parameter $s$ (days)$^{-1}$ of the continuum model. We estimated the doubling time from the exponential phase of growth, $e^{st}$, and the resulting formula, $t_{\text{double}} = \ln2/s$. Considering only the doubling time would cause the cells to divide infinitely, which is not consistent with the logistic growth dynamics. Hence, we introduced the effect of space availability, as well as a spontaneous death probability that reduced cell viability (Section 2 of S1 File). The probability of spontaneous death was tuned using data obtained from our cell viability assay using flow cytometry (Section 3 of S1 File). For the treatment conditions, the cell death rate $k$ was translated to a cell death probability $P_k = k dt$ for each time step of the simulation. The cell-division mechanism algorithm separates into two processes based on the cell position in space. If the cell is attached to the glass and there is sufficient space, then division will be performed on the glass; otherwise, the cell divides in any direction of the 3D space if there is sufficient space. However, if there is insufficient space, the cell becomes quiescent. If the cell is not ready to divide, CA turns into a migration mechanism.

The first condition for migration considered an adhesion parameter, and the second is the state of the cell. The adhesion parameter is the local density, defined as the sum of the densities in the neighbouring positions, i.e., $\sum_{n=\{-3,3\}}(u_{i+n,j,k} + u_{i,j+n,k} + u_{i,j,k+n})$. A cell can migrate if the local density is equal to or greater than the threshold value $n$, which is related to the number of neighbouring cells. We hypothesized that the number of neighbours required for cell migration would increase over time (Fig H b of S1 File) because the initial cell distribution in 3D space was sparse, hence they could migrate freely to search for other cells to attach. However,

as cell clustering occurs due to cell division or cell contact, migration becomes less frequent as the cells become more attached to each other. If a cell satisfies these conditions, the algorithm checks its position. If a cell is settled at the bottom of the space or is connected to a cell located at the bottom, it cannot migrate; otherwise, the cell can migrate in the 3D space given the moving probabilities $P_0, \ldots, P_6$. The hybrid model was implemented in CUDA-C/C++ language [95] and the code is uploaded to the corresponding GitHub repository.

### 5.9 Continuum model calibration

The model, $M$, (Eqs (1)–(3)) includes a set of parameters $\theta = \{D_u, s, \chi, D_f, r\}$ that are considered unknown. We used their Probability Density Functions (PDF) and the calculated densities from the 3D cell culture data, $\mathscr{D}$, to assess the most probable parameter values according to Bayes' rule

$$\mathbb{P}(\theta|\mathscr{D}, M) \propto \mathbb{P}(\mathscr{D}|\theta, M)\mathbb{P}(\theta) \tag{8}$$

where $\mathbb{P}(\theta|\mathscr{D}, M)$ is the posterior PDF of the model parameters $\theta$ given the observed data $\mathscr{D}$ and the model $M$, $\mathbb{P}(\mathscr{D}|\theta, M)$ is the likelihood of the observed data $\mathscr{D}$ given the model $M$ and the parameters $\theta$, and $\mathbb{P}(\theta)$ is the prior PDF. We assume uninformative, uniform distributions for the model parameter prior PDFs.

The experimental data consisted of 12 control, and 5 datasets per treatment dose (4 doses). Each dataset consisted of samples collected at 7 time-points for controls and 5 time-points for the treatment data. The datasets were assumed to be independent, and the model was evaluated separately for each dataset. The likelihood was defined as

$$L(\theta; \mathbf{d}) = \prod_{i=1}^{n} \frac{1}{\sigma_d \sqrt{2\pi}} \exp\left(-\frac{(d_i - q_i(\theta))^2}{2\sigma_d^2}\right) \tag{9}$$

where $n$ is the number of spatial grid points, $\mathbf{d}$ the density profile of the corresponding sample in a dataset, $d_i$, $q_i$ the density values of the experimental sample and the simulation results, respectively, at grid point $i$, and $\sigma_d$ is the variance of the distribution of the likelihood.

We used a Transitional Markov Chain Monte Carlo (TMCMC) [96–98] algorithm implemented in the Π4U package [99]. The TMCMC algorithm iteratively constructs series of intermediate posterior PDFs

$$\mathbb{P}_j(\theta|\mathscr{D}, M) \propto \mathbb{P}(\mathscr{D}|\theta, M)^{\rho_j}\mathbb{P}(\theta) \tag{10}$$

where $j = 0, \ldots, m$ is the index of the Monte Carlo time series (generation index), $\rho_j$ controls the transition between generations, and $0 < \rho_0 < \rho_1 < \cdots < \rho_m = 1$. The intermediate PDFs converge to the target PDF as generations progress [96] (Section 3 of S1 File). The TMCMC method can utilize a large number of parallel chains that are evaluated in each Monte Carlo step to obtain a result close to the true posterior PDF. To evaluate the calibration results, we calculated the Normalized Root Mean Squared Error (NRMSE) (Section 4 of S1 File). As described in our previous work [49], we used all the time points for the calibration of the continuum model. Validation was performed using the hybrid (discrete-continuum) model with the spatial statistical measures described below. The code for the calibration is available in the corresponding GitHub repository.

### 5.10 Spatial analysis

**5.10.1 Complete spatial randomness test of spatial cell distributions.** The Complete Spatial Randomness (CSR) test examines whether the observed spatial point patterns, in our

case the centroids of the nuclei, can be described by a uniform random distribution [100]. The CSR test was implemented using Ripley's $K$-function and the *spatstat* [101] package in R [102]. The $K$-function [103] is defined as the ratio between the number of the events, i.e. locations of points, $j$ within a distance $t$ from the event $i$, over the total number of events $N$, in the studied volume $V$

$$K(t) = \frac{1}{\hat{\lambda}} \sum_i \sum_{j \neq i} I(d_{ij} < t), \ I(x) = \begin{cases} 1, & \text{if } x = \text{true} \\ 0, & \text{otherwise} \end{cases} \tag{11}$$

where $\hat{\lambda} = N/V$ denotes the average density of events $N$ in the studied volume $V$, $d_{ij}$ is the distance between events $i$ and $j$, and $t$ is the search radius. The $K$-function was calculated for all datasets and compared against complete spatial randomness following a Poisson process $K(t) = 4\pi t^3/3$ [103] for three spatial dimensions. Isotropic edge correction was applied in the calculation of the $K$-function. The volume used for the calculation was the same as that used in the simulations, i.e., $2.5 \times 2.5 \times 0.917$ mm$^3$. To assess the uncertainty of the random variable $K$, we produced a CSR envelope by generating 100 random distributions and calculating the $K$-function for each distribution. The envelope was created by keeping the minimum and maximum values of the resulting $K$ values. A substantial upward separation of the observed $K$-function from the theoretical random $K$-function denotes clustered patterns, and downward separation denotes dispersed patterns [100]. Both separation types suggest non-randomness of the examined spatial distributions.

**5.10.2 Characterization of the Spatial Cell Distributions.** The *inter-nucleic (IN) Distance Distribution* for a given sample was calculated using pairwise Euclidean distances between all nuclei. Given two nuclei $i$ and $j$ with centroid positions $\mathbf{p_i} = (x_i, y_i, z_i)$ and $\mathbf{p_j} = (x_j, y_j, z_j)$ respectively, their pairwise Euclidean distance is given by

$$D_{ij} = \sqrt{(x_i - x_j)^2 + (y_i - y_j)^2 + (z_i - z_j)^2}, \ i,j = 1 \dots N, \ i \neq j \tag{12}$$

where $N$ denotes the total number of nuclei.

The *Nearest-Neighbour (NN) Distance Distribution* for a given sample was calculated using the distances between the nearest neighbours of the nuclei. The nearest neighbour distance for a given nucleus $i$ is given by the minimum IN Distance between the nucleus $i$ and all the other nuclei of the sample, i.e. $D_{NN}^i = \min_{i,j} \{D_{ij}\}, j \in [1, N], j \neq i$.

Comparisons between *in vitro* and *in silico* IN and NN distance distributions were performed using the cosine similarity test [104] in MATLAB [77]. The code for the spatial analysis is available in the corresponding GitHub repository.

## Supporting information

**S1 File. Supporting information file.**
(PDF)

## Acknowledgments

NMD thanks Rémi Dagenais (McGill) for the useful discussions regarding the article.

## Author Contributions

**Conceptualization:** Nikolaos M. Dimitriou, Georgios D. Mitsis.

**Data curation:** Nikolaos M. Dimitriou, Maria Kyriakidou.

**Formal analysis:** Nikolaos M. Dimitriou.

**Funding acquisition:** Nikolaos M. Dimitriou.

**Investigation:** Nikolaos M. Dimitriou.

**Methodology:** Nikolaos M. Dimitriou, Salvador Flores-Torres.

**Project administration:** Georgios D. Mitsis.

**Resources:** Salvador Flores-Torres, Joseph Matthew Kinsella, Georgios D. Mitsis.

**Software:** Nikolaos M. Dimitriou.

**Supervision:** Joseph Matthew Kinsella, Georgios D. Mitsis.

**Validation:** Nikolaos M. Dimitriou.

**Visualization:** Nikolaos M. Dimitriou.

**Writing – original draft:** Nikolaos M. Dimitriou.

**Writing – review & editing:** Nikolaos M. Dimitriou, Salvador Flores-Torres, Maria Kyriaki-dou, Joseph Matthew Kinsella, Georgios D. Mitsis.

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
