## [Decision Letter · Decision Letter 0]

13 Sep 2023

Dear Mr Dimitriou,

Thank you very much for submitting your manuscript "Cancer cell sedimentation in 3D cultures reveals active migration regulated by self-generated gradients and adhesion sites" for consideration at PLOS Computational Biology.

As with all papers reviewed by the journal, your manuscript was reviewed by members of the editorial board and by several independent reviewers. The reviewers (please see below this email) all agree that this is an important piece of work, but they do highlight a number of major concerns. We would therefore like to invite the resubmission of a significantly-revised version that takes into account the reviewers' comments.

We cannot make any decision about publication until we have seen the revised manuscript and your response to the reviewers' comments. Your revised manuscript is also likely to be sent to reviewers for further evaluation.

Sincerely,

Philip K Maini

Academic Editor

PLOS Computational Biology

Daniel Beard

Section Editor

PLOS Computational Biology

Reviewer's Responses to Questions

**Comments to the Authors:**

Reviewer #1: Please refer to the attached PDF.

Reviewer #2: This is a well written manuscript combining experiments with mathematical modeling for a better understanding of the sedimentation of cancer cells in 3D cultures. The article contains nice and novel experimental findings, which would make the article suitable even for interdisciplinary journals, beyond specialized computational biology journals. The mathematical models employed are not novel but their combination with the experimental data to provide a better understanding of the mechanisms of cancer cell sedimentation is novel and interesting. I have some specific comments to make

1. A paragraph in the Methods describing the overall strategy of the study would be helpful. The authors provide briefly a description in the beginning of the Results and throughout the presentation of the results but it is not sufficient. I would suggest to add a separate paragraph in the Methods describing the strategy of the study.

2. The mathematical models are not clearly presented. A schematic of the models along with any boundary and initial conditions is missing. Also a clear presentation of the values of model parameters is missing. There is a reference to what the authors found to be the most critical parameters but a detailed presentation in form of Table(s) showing the parameters, their values, and references from where these values were taken or how they were defined is missing.

3. I really like figure 1 but the fonts in panels c and d are too small to read them. The same applies to panels of Figures 3 and 4.

4. The Discussion is lengthy and repeats the Results. I think it should be shortened significantly and be a discussion of issues that have been discussed previously in the manuscript.

Other than these issues, I think it is a good work to be published in PLOS Comp. Biology.

Reviewer #3: Review report of ``Cancer cell sedimentation in 3D cultures reveals active migration regulated by self-generated gradients and adhesion sites'' by NM Dimitiou, S Flores-Torres, M Kyriakidou, J Kinsella, and GD Mitsis

The authors study the phenomena of cell sedimentation in which cells tend to undergo downward vertical migration in 3D hydrogel cultures. The authors proposed a model based on active migration based on a chemotactic gradient that is self-generated by the cells. They calibrate a mathematical model using cell culture data and conclude that this provides evidence for their theory.

There are a few aspects of the work that need some clear explanation before I can recommend publication.

Major comments:

1. Why is understanding cell sedimentation important? The only motivation I can find in Lines 50-63 which essential reduce it to "generally undesirable" and typically assume to be gravity based "least resistance". This does not convince me that this is an important problem. I am not saying it isn't, but the authors should state their case more strongly.

2. The other thing I noted is that there does not seem to be any model selection performed and the main evidence against gravity induced migrations is not based on the any underlying model (Section 3A). It concerns me a little that Paclitaxel arrests both active migration and proliferation... could it not be possible that if proliferation was present that sedimentation could be observed? To me it would be desirable to calibrate both the model presented in Section 2H with an alternative representing the gravity induced migration (e.g., no chemotaxis but motility bias base on gravity). The number of parameters will differ in each model so some model selection measure will be required.

3. In the model calibration I do not see the details on the priors, these need to be reported and some exploration of sensitivity to the priors are needed.

4. The paper is quite difficult to read as it is, it feels it was written to be read as Intro-> Results -> Discussion with Methods as an Appendix. This makes it quite hard to unpack the methods if reading in order. Id suggest restructure, or give the reader more signposts.

Minor comments:

1. Line 168-169 I don't see how Eq (2) is a signal production term... it looks like diffusion of chemotactic agent to me.

2. I believe TMCMC is a special case of the widely used Sequential Monte Carlo. I would expect the authors to include references to the key literature in that space.

**Have the authors made all data and (if applicable) computational code underlying the findings in their manuscript fully available?**

Reviewer #1: Yes

Reviewer #2: None

Reviewer #3: Yes

PLOS authors have the option to publish the peer review history of their article (what does this mean?). If published, this will include your full peer review and any attached files.

Reviewer #1: No

Reviewer #2: No

Reviewer #3: No
---

## [Decision Letter · Decision Letter 1]

26 Mar 2024

Dear Mr Dimitriou,

Thank you very much for submitting your manuscript "Cancer cell sedimentation in 3D cultures reveals active migration regulated by self-generated gradients and adhesion sites" for consideration at PLOS Computational Biology. As with all papers reviewed by the journal, your manuscript was reviewed by members of the editorial board and by several independent reviewers. The reviewers appreciated the attention to an important topic. Based on the reviews, we are likely to accept this manuscript for publication, providing that you modify the manuscript according to the review recommendations.

Sincerely,

Philip K Maini

Academic Editor

PLOS Computational Biology

Daniel Beard

Section Editor

PLOS Computational Biology

Reviewer's Responses to Questions

**Comments to the Authors:**

Reviewer #1: This manuscript describes an integrated experimental/theoretical study investigating the mechanisms behind 3D cell sedimentation within hydrogels. The authors expose breast cancer cells to Paclitaxel, an inhibitor of microtubule assembly, to argue that the largely vertical migration of cells is due to cell-generated forces rather than gravity or ECM compression. Bulk RNA sequencing of cells in control and treatment settings further suggest that this anisotropic movement relies on pathways related to collective behavior. From these observations, the authors hypothesize that such movement arises from chemotactic leader-follower processes such as those seen during wound healing. In this scenario, cells at the bottom of the gel would adhere to the underlying glass substrate and secrete a diffusing chemical signal that attracts clusters of cells near the top of the assay. The authors examine this idea within a theoretical setting by fitting and simulating a hybrid mathematical model that is similar to those used to describe wound healing. The authors find that their in silico framework yields results similar to those observed in vitro, both in terms of the nearest neighbor distributions and more sophisticated spatial metrics such as Ripley’s K-function. The authors conclude by performing a series of in vitro experiments that aim to corroborate their chemotactic hypothesis. These include tests in which cells are seeded in a less adhesive agarose gel, in addition to those in which cancer cells are co-cultured with fibroblasts. In these latter two experiments, cells do not travel as far as they do in Matrigel.

This paper will be acceptable for publication pending some minor revisions (see below). The paper provides a good example of how mathematical modeling can be combined with experiments to produce some new insights. The authors have addressed many of my major issues on their previous iteration of their manuscript: for instance, in their new version it is much clearer why the authors utilize a model incorporating chemotaxis to a diffusing chemical signal and why they disregard other mechanisms such as gravity. Additionally, the importance of their work to the scientific community is made much clearer. While this model only examines one possible explanation for the sedimentation behavior, the authors make it clear in the discussion that there are potentially other mechanisms that may play a role and which they cannot rule out at this time. The paper is well written, and this reviewer found the presentation of their experiments in the second draft much more understandable than in the original version.

Below please find a list of minor comments that should be addressed.

List of Minor Comments:

1.   Figure 3a: The large range of identified values for some of the parameters suggests that the mathematical model that the authors are using is not identifiable (that is, there may be more than one set of parameter values that can accurately capture the observed data). Can the authors comment on this possibility in the main text?

2.   Abstract: If there is space available, it may be helpful to introduce the importance of this problem in the abstract.

3.   Line 179: It would be helpful to list the biological interpretation of the parameter s, similar to the sentence previous to this.

4.   Line 232-233: Do the authors have results that confirm that fibroblasts upregulate their own protein / chemical secretion? Could another explanation be that cells adhere to the fibroblasts or are prevented from moving down due to overcrowding? Do the fibroblasts move from their original positions?

5.   Line 256: Can the authors cite a reference for this fact about matrigels?

6.   Line 290: Do the authors mean in this sentence that contact of inhibition may also be present, based on the expression profiles in the treated and control samples?

7.  References 13, 41, 53, 91, 94: The link / DOI given in this section does not work.

8.  Supplementary Figure S.14: It may be helpful to state in the caption what parameter values correspond to the low diffusion / high advection and high diffusion / low advection states.

Reviewer #2: I recommend the manuscript for publication as it is.

Reviewer #3: I Thanks the authors considered responses to my initial comments. I have no other concerns, perhaps a small addition to the manuscript about why no model selection was performed would be useful to a reader (the authors' response to my comment 2 should be reflected somewhere in the main manuscript). Computational limitations are perfectly legitimate reasons.

**Have the authors made all data and (if applicable) computational code underlying the findings in their manuscript fully available?**

Reviewer #1: Yes

Reviewer #2: Yes

Reviewer #3: Yes

PLOS authors have the option to publish the peer review history of their article (what does this mean?). If published, this will include your full peer review and any attached files.

Reviewer #1: No

Reviewer #2: No

Reviewer #3: No

Figure Files:

Data Requirements:

Reproducibility:

References:

---

## [Decision Letter · Decision Letter 2]

25 Apr 2024

Dear Mr Dimitriou,

We are pleased to inform you that your manuscript 'Cancer cell sedimentation in 3D cultures reveals active migration regulated by self-generated gradients and adhesion sites' has been provisionally accepted for publication in PLOS Computational Biology.

Best regards,

Philip K Maini

Academic Editor

PLOS Computational Biology

Daniel Beard

Section Editor

PLOS Computational Biology

Reviewer's Responses to Questions

**Comments to the Authors:**

Reviewer #1: I thank the authors for again taking the time to address my points. I have no other concerns.

**Have the authors made all data and (if applicable) computational code underlying the findings in their manuscript fully available?**

Reviewer #1: Yes

PLOS authors have the option to publish the peer review history of their article (what does this mean?). If published, this will include your full peer review and any attached files.

Reviewer #1: No

---

## [Editor Report · Acceptance letter]

6 Jun 2024

PCOMPBIOL-D-23-00758R2 

Cancer cell sedimentation in 3D cultures reveals active migration regulated by self-generated gradients and adhesion sites

Dear Dr Dimitriou,

I am pleased to inform you that your manuscript has been formally accepted for publication in PLOS Computational Biology. Your manuscript is now with our production department and you will be notified of the publication date in due course.

With kind regards,

Lilla Horvath
